# Is There a FADS2-Modulated Link between Long-Chain Polyunsaturated Fatty Acids in Plasma Phospholipids and Polyphenol Intake in Adult Subjects Who Are Overweight?

**DOI:** 10.3390/nu13020296

**Published:** 2021-01-21

**Authors:** Manja M. Zec, Irena Krga, Ljiljana Stojković, Maja Živković, Biljana Pokimica, Aleksandra Stanković, Maria Glibetic

**Affiliations:** 1Center of Research Excellence in Nutrition and Metabolism, Institute for Medical Research, National Institute of the Republic of Serbia, University of Belgrade, 11000 Belgrade, Serbia; irenakrga@yahoo.com (I.K.); biljana.pokimica@hotmail.com (B.P.); mglibetic@gmail.com (M.G.); 2Laboratory for Radiobiology and Molecular Genetics, Vinča Institute of Nuclear Sciences, National Institute of the Republic of Serbia, University of Belgrade, 11000 Belgrade, Serbia; ljiljanas@vinca.rs (L.S.); majaz@vinca.rs (M.Ž.); alexas@vinca.rs (A.S.)

**Keywords:** polyphenols, Aronia, long-chain polyunsaturated fatty acids, *FADS2*, obesity, gene-dietary interaction

## Abstract

Dietary polyphenols promote cardiometabolic health and are linked with long-chain polyunsaturated fatty acids in plasma phospholipids (LC-PUFA). The *FADS2* polymorphisms are associated with LC-PUFA metabolism and overweight/obesity. This 4-week study examined the link between polyphenol intake, *FADS2* variants (rs174593, rs174616, rs174576) and obesity in 62 overweight adults (BMI ≥ 25), allocated to consume 100 mL daily of either: Aronia juice, a rich source of polyphenols, with 1177.11 mg polyphenols (expressed as gallic acid equivalents)/100 mL (AJ, *n* = 22), Aronia juice with 294.28 mg polyphenols/100 mL (MJ, *n* = 20), or nutritionally matched polyphenol-lacking placebo as a control (PLB, *n* = 20). We analyzed LC-PUFA (% of total pool) by gas chromatography and *FADS2* variants by real-time PCR. Four-week changes in LC-PUFA, BMI, and body weight were included in statistical models, controlling for gender and PUFA intake. Only upon AJ and MJ, the presence of *FADS2* variant alleles affected changes in linoleic, arachidonic, and eicosapentaenoic acid (EPA). Upon MJ treatment, changes in EPA were inversely linked with changes in BMI (β= −0.73, *p* = 0.029) and weight gain (β= −2.17, *p* = 0.024). Only in subjects drinking AJ, the link between changes in EPA and anthropometric indices was modified by the rs174576 variant allele. Our results indicate the interaction between *FADS2*, fatty acid metabolism, and polyphenol intake in overweight subjects.

## 1. Introduction

According to the 2016 estimates, more than 1.9 billion adults representing 39% of the world’s adult population are overweight, and of these, around 650 million are obese [1]. Overweight and obesity are defined as excess fat accumulation that poses a risk to an individual’s health, with body mass index (BMI) greater or equal to 25 kg/m^2^ indicating overweight and 30 kg/m^2^ or higher representing obesity [1]. These conditions are linked with dyslipidemia, insulin resistance, and hypertension and are a major risk factor for developing inflammation-associated cardiovascular diseases (CVD) and type 2 diabetes as well as some types of cancer, having a high impact on morbidity and mortality [1,2].

Fats are the principal high-energy food components, and it is generally accepted that their increased consumption combined with physical inactivity and genetic background might be associated with the overweight/obesity pandemic. Fatty acid metabolism is of critical importance for the utilization of dietary fats for different physiological processes in the body, and altered plasma fatty acid profiles, including changes in long-chain (LC-) polyunsaturated fatty acids (PUFA), have been associated with different diet-related disorders such as for overweight and obesity [3,4]. Aside from diet, LC-PUFA can be obtained endogenously by metabolic conversion of their dietary precursors α-linolenic acid (C18:3*n* − 3, αLNA) and linoleic acid (LA, C18:2*n* − 6)—the essential PUFA from the omega-3 (*n* − 3) and omega-6 (*n* − 6) families that humans cannot synthesize. The αLNA is a precursor for eicosapentaenoic acid (EPA, C20:5*n* − 3) and docosahexaenoic acid (DHA, C22:6*n* − 3) with reported anti-inflammatory, anti-thrombotic, and blood pressure-lowering effects [5]; while linoleic acid (LA, C18:2*n* − 6) is metabolized to arachidonic acid (AA, C20:4*n* − 6) that serves as a precursor for several proinflammatory and aggregatory mediators [6]. The LC-PUFA endogenous synthesis involves a series of alternating desaturation and elongation steps shared between *n* − 3 and *n* − 6 metabolic pathways. Delta-5 and delta-6 desaturases are the rate-limiting enzymes in LC-PUFA production encoded by the fatty acid desaturase 1 (*FADS1)* and *FADS2* genes, respectively [7]. Single nucleotide polymorphisms (SNPs) in *FADS* genes are with race- and ethnic-specific distributions and associated with levels of circulating LC-PUFA [8,9,10], and several SNPs in *FADS1*/*FADS2* have been linked with a risk of overweight and obesity [7,11,12]. Thus, genetic variants in the *FADS* gene cluster might alter fatty acid profiles, thereby modulating fat metabolism and influencing the development of different diet-related disorders.

Polyphenols are secondary plant metabolites, and accumulating evidence suggests their beneficial effects on different aspects of cardiometabolic health. These phytochemicals are widely recognized for their anti-inflammatory activities, but also exert antioxidant, anti-thrombotic, antimicrobial, and other biological effects [13,14]. Polyphenols are substantial components of the human diet, with fruits, vegetables, and plant derived-beverages as main dietary sources [13], with *Aronia melanocarpa*, among other berries, containing substantial amounts of polyphenols linked with beneficial CVD effects [15,16]. Numerous epidemiological studies have linked higher polyphenol intakes with a reduced risk of chronic diseases including CVD and type 2 diabetes, as well as CVD-related mortality [17,18,19]. Cardiometabolic benefits of dietary polyphenols are also supported by clinical and experimental data indicating their positive influence on fatty acid metabolism, blood lipid profiles, blood pressure, blood glucose, and endothelial and platelet function [16,20,21,22,23,24]. Despite this large body of evidence for polyphenol effects, only a limited number of studies have examined the possible interaction between polyphenol intake and genetic polymorphisms on different cardiometabolic biomarkers [25,26,27]. Furthermore, even less is known about the interaction of polyphenols and genes concerning genetic regulation of endogenous fatty acid metabolism and obesity-related traits.

A precision nutrition study from our laboratory demonstrated associations between variant allele presence in *FADS2* (rs174593, rs174616, rs174576) and AA and desaturase-5 activity in adult subjects [28]. The same study observed independent associations between BMI and predicted activity of desaturase-5 enzyme regardless of the *FADS2* genetic structure [28]. Furthermore, in a sample of apparently healthy adults, we demonstrated dietary effects of polyphenols from *Aronia melanocarpa* (chokeberry) juice on fatty acid profiles in plasma phospholipids as seen in increased saturated and decreased PUFA, the latter due to reduced LA, upon 4-week polyphenol intervention [29]. However, the same study failed to demonstrate an interventional effect of dietary polyphenols on obesity parameters [29], and the relationship between polyphenol intake, fatty acid metabolism, and obesity remains unclear. Therefore, the primary objective of this study was to examine whether there is a link between LC-PUFAs in plasma phospholipids and measures of obesity (BMI and weight gain) in subjects who are overweight; and whether the relationship is modulated by the 4-week daily intake of polyphenol-rich Aronia juice. Furthermore, we evaluated whether the relationships depend on the *FADS2* genetic variants.

## 2. Materials and Methods

### 2.1. Ethical Clearance, Study Design and Intervention Treatments

The present work forms part of a larger randomized, 6-month, three-arm, placebo-controlled interventional study, registered at ClinicalTrials.gov under identifier NCT02800967, assessing the effects of long-term intake of the polyphenol-rich Aronia juice. The original study was a cross-over and included three 4-week interventional periods. For this study, a subgroup of subjects with baseline BMI ≥ 25 kg/m^2^ were selected and a total of 62 subjects (BMI (mean ± SD) = 30.11 ± 4.35 kg/m^2^), who took part in the first 4-week interventional period, were included in the current analysis. Thus, the current study design presented a 4-week, parallel, three-arm, placebo-controlled, interventional study in subjects who are overweight.

The original study included men and women, aged between 28 and 56 years, at low CVD risk defined as the presence of at least one of the following: BMI ≥ 25 kg/m^2^, central obesity (waist circumference ≥80 cm for women and ≥94 cm for men) and blood pressure above optimal values (systolic/diastolic blood pressure > 120/80 mg Hg). Exclusion criteria included clinical signs and symptoms of CVD and other chronic or acute diseases, smoking, pregnancy, lactation, use of medications, supplements, allergy or intolerance to berries or Aronia juice components, and blood donations within the previous 4 months (Appendix A).

Recruited subjects were randomly assigned to consume during the four weeks a 100 mL daily dose of either of the three interventional treatments: (1) Aronia juice containing 1177.11 mg polyphenols/100 mL (AJ), (2) Aronia juice containing 294.28 mg polyphenols/100 mL (MJ), and (3) polyphenol-free placebo (PLB). The subjects consumed their daily dose after breakfast. The AJ was certified as a dietary supplement with the Ministry of Health of the Republic of Serbia and provided by Nutrika LTD (Belgrade, Serbia). The PLB was safe for human consumption, lacked polyphenols, and was macro- and micronutrient matched to AJ, with the same color, taste, smell, and texture [30]. The MJ was prepared as a 25% (*v*/*v*) solution of AJ in PLB. Total polyphenol contents of AJ and MJ were previously estimated by the Folin–Ciocalteu method and expressed as gallic acid equivalents [31]. A detailed composition of the AJ is available elsewhere [30]. During the study, subjects were asked to maintain their regular dietary habits and physical activity levels but were instructed to refrain from berries and berry-derived products and avoid excess amounts of other polyphenol rich-foods like green-tea, olive oil, and nuts. Study treatments were provided in dark bottles and participants were instructed to keep them in the refrigerator to preserve the stability of anthocyanins. Compliance with treatment was evaluated by collecting empty bottles on the last day of the intervention.

All participants were fully informed about the study protocols and signed informed consent prior to their enrollment in the interventional phase. The study was conducted following the Helsinki Declaration and all procedures were approved by the Ethics Committee of the Clinical Hospital Center in Zemun (Belgrade, Serbia; ref. no. 2125/2013).

### 2.2. Dietary Intake and Anthropometric Data Assessments

Dietary intake data were assessed from repeated 24-hour dietary recalls for two non-consecutive days obtained during the structured interviews with trained researchers. To increase the accuracy of reporting portion size, subjects were provided with a photo-booklet containing reference portions for 125 items including simple foods and regularly consumed composite dishes. Dietary data were analyzed by Diet Assess & Plan platform for standardized food consumption data collection and comprehensive nutritional assessment [32]. Total energy and intakes of main nutrients (carbohydrates, protein, and fats), main types of fatty acids (saturated, monosaturated, and PUFA), and individual fatty acids were calculated using the Serbian Food Composition Database, developed according to EuroFIR standards [28,33]. This database was also employed to determine polyphenol intake using the data from the online free database Phenol-Explorer (http://phenol-explorer.eu).

Anthropometric parameters were measured at the baseline and after a 4-week intervention period following the standard protocols [29]. Bodyweight was assessed using the TANITA UM072 bioelectrical impedance analyzer (Hong Kong, Japan), height with a height measuring scale, and waist and hip circumference with a non-elastic flexible tape measure. The BMI was calculated by dividing body weight in kilograms with the square of height in meters.

### 2.3. Blood Sample Collection and Processing

Blood sampling was performed in the morning after an overnight fast (10–12 h), at baseline, and the end of the 4-week study (Appendix A). Venous blood was collected into serum and ethylenediaminetetraacetic acid (EDTA)-coated tubes and centrifuged at 2500 rpm for 5 min to obtain serum and plasma samples, respectively. Biochemical parameters (lipid status, glucose, liver enzyme activities, urea, creatinine, uric acid, and lactate dehydrogenase) were determined in fresh serum samples using the Cobas c111 clinical biochemistry analyzer and reagent kits (Roche, Basel, Switzerland), following the manufacturer’s instructions. Plasma samples were kept at −80 °C until assayed. Additionally, fresh whole blood samples collected into EDTA-coated tubes were used for genomic DNA extraction and genotyping.

### 2.4. Plasma Phospholipid Fatty Acid Analysis

Total lipids were extracted from plasma samples according to the Folch method [34], using the chloroform/methanol mixture (2:1, *v*/*v*) with 0.05% (*w*/*v*) butylated hydroxytoluene as an antioxidant as described before [28]. Plasma phospholipids were further isolated from other lipid subclasses by one-dimensional thin-layer chromatography on silica gel coated plates (Merck, Darmstadt, Germany) using petroleum ether: diethyl ether: acetic acid solvent system (87:12:1, *v*/*v*/*v*). Afterward, fatty acid methyl esters were prepared by transmethylation with 2M sodium hydroxide in methanol for 1 h at 85 °C, and 1M sulfuric acid in methanol for 2 h at 85 °C [29]. Samples were cooled to room temperature, centrifuged at 3000 rpm for 10 min, and subsequently, the upper phase was collected and dried under a gentle nitrogen stream. Fatty acid methyl esters were reconstituted in hexane prior to analysis and separated with the RTX 2330 capillary column (60 m × 0.25 mm × 0.2 μm, RESTEK, Bellefonte, PA, USA) using the Shimadzu GC-2014 gas chromatograph (Kyoto, Japan) with flame ionization detector. The temperature of the injector was set at 220 °C and the detector at 260 °C. The column temperature settings were as follows: 140 °C for 5 min, increase to 220 °C at the rate of 3 °C/min, and 220 °C for 20 min. The flow of carrier gas (helium), air, and hydrogen were 5 mL/min, 320 mL/min, and 30 mL/min, respectively. Fatty acids were identified by comparing peak retention times with standard mixtures (PUFA-2 and 37 Component FAME Mix, Supelco Inc., Bellefonte, PA, USA). 

For this study, we evaluated the following fatty acids: palmitic acid (C16:0), palmitoleic acid (C16:1*n* − 7), stearic acid (C18:0), vaccenic acid (C18:1*n* − 7), oleic acid (C18:1*n* − 9), LA, dihomo-γ-linolenic acid (DGLA, C20:3*n*− 6), AA, adrenic acid (C22:4*n* − 6), αLNA, EPA, clupanodonic acid (C22:5*n* − 3), and DHA. The individual fatty acid content was expressed as a percentage of the total pool of identified fatty acids in plasma phospholipids. Activities of enzymes involved in the synthesis of long-chain fatty acid products were estimated using the fatty acid product-to-precursor ratios: AA/DGLA for delta-5 desaturase activity and DGLA/LA for delta-6 desaturase activity.

### 2.5. Genotyping

Analyzed SNPs within *FADS2* included rs174576 (intron 1), rs174593 (intron 5), and rs174616 (intron 7), selected based on position, functional context and evidence-based data on SNP significance, as previously described [28].

The DNA was isolated from fresh whole blood samples drawn in EDTA-coated tubes using phenol-chloroform extraction [35]. Selected *FADS2* variants were determined using TaqMan^®^ Assays for SNP genotyping, designed, and functionally tested by the manufacturer, Thermo Fisher Scientific. The context sequence, within which the allele-specific probes are designed, and other assay details are provided online (https://www.thermofisher.com) for each assay used, under its product name: C___2575520_10 (for rs174576 polymorphism), C___2575513_10 (for rs174593 polymorphism), and C___2268923_10 (for rs174616 polymorphism). The PCR and genotype determination were performed using the 7500 Real-Time PCR System with SDS software version 1.4.0 (Applied Biosystems Inc., Foster City, CA, USA).

### 2.6. Statistical Analyses

Statistical analyses were performed by SPSS ver. 24 (Chicago, IL, USA), and a probability threshold of 0.05 was relevant. For graphical representation, we used the SPSS and Microsoft Excel 2016. Depending on the distribution, the data are presented as means ± SEM for normally distributed, otherwise median (interquartile range). We applied general linear model and Kruskal–Wallis analyses, for parametric and non-parametric approach, respectively; to test baseline differences between parameter distributions across the study treatments with varying polyphenol content, and furthermore, to test the differences in changes of the parameters across the interventional groups, and genotypes in a dominant model. We tested the deviation from Hardy–Weinberg equilibrium, by using χ^2^ test with 1 df, with probability threshold set at 0.05. Missing data were completely at random and we included those in the analyses by intention-to-treat principle, accounted for in statistical syntax.

Our variables included changes in the anthropometric indices (BMI and total body weight) and LC-PUFA in plasma phospholipids (LA, AA, adrenic acid, EPA, clupanodonic acid, DHA, and predicted desaturase activities) upon 4-week interventional treatments with varying polyphenol content. To test whether the variables depend on the *FADS2* genotype in an additive model, we applied hierarchical multiple regressions to evaluate the contribution of the *FADS2* variant alleles (within rs174593, rs174616, and rs174576) to the variability of the changes, and constructed crude and multivariable models, the latter fully adjusted for gender and total daily *n* − 6/*n* − 3 intakes.

To assess the main study objective, we constructed similar multivariable-adjusted regression models to test whether the variability in the changes in the anthropometric variables upon the 4-week polyphenol treatment, depends on the variability in changes in the LC-PUFA. To examine whether the polyphenol-rich Aronia juice affected the associations, the regressions were further tested for the effect modification by the interventional treatments with varying polyphenol content.

Finally, to test if there was a gene-dietary interaction between polyphenol intake and *FADS2* variant alleles if upon a multivariable regression analyses changes in the LC-PUFA were significantly associated with the changes in anthropometric indices, we included an interaction term to test whether the presence of a variant allele within the selected *FADS2* polymorphisms interacted with the associations.

## 3. Results

### 3.1. Clinical Characteristics and FADS2 Genotypes of the Study Subjects Who Are Overweight

Our study included 62 subjects who were overweight (BMI ≥ 25), mean age = 41.05 ± 0.85, and out of which 54.8% were women. Baseline characteristics across the allocated treatments are presented in Table A1. All of the parameters were homogenously distributed, except for the glucose, which was random with higher levels in subjects allocated to PLB (*p* = 0.022). Intake of individual dietary fatty acids and levels of individual fatty acids in plasma phospholipids were balanced across the interventional treatments (Appendix A). Genotype frequencies were in line with Hardy–Weinberg equilibrium and in line with previously reported (Appendix A [28]).

### 3.2. Effects of the 4-Week Interventional Treatment with Varying Polyphenol Content on Changes in BMI, Total Body Weight and Levels of LC-PUFA in Plasma Phospholipids in the Subjects Who Are Overweight

Changes in the anthropometric parameters and the levels of individual LC-PUFA in the plasma phospholipids across the interventional treatments with varying polyphenol content are presented in Figure 1. In comparison with PLB, the polyphenol-containing treatments tended to result in a lower increase in AA (*p* = 0.070, Figure 1a). Although failing to reach statistical significance, it is worth noting that the 4-week AJ treatment with considerably high polyphenol content, exhibited the indicative potential in preventing weight gain, as seen in changes in BMI (median [IQR]: −0.09 [−0.44, 0.09]) and body weight (median [IQR]: −0.25 [−1.25, 0.25]) (Figure 1a). In addition, polyphenol-containing treatments virtually induced a dose-dependent increase in changes in the levels of EPA and DHA (Figure 1a), although failing to reach a statistical significance (*p* = 0.736 and 0.172, respectively). Furthermore, we observed significantly stronger effects on AA/DHA decrease with the AJ treatment (*p* = 0.021, Figure 1b), with no significant effects on AA/EPA and predicted desaturase 5 or 6 activities.

### 3.3. Relationship between Dietary Intake of Polyphenols from Aronia Juice, FADS2 Genotypes, Levels of LC-PUFA in Plasma Phospholipids and Obesity Parameters

Changes in anthropometric parameters and LC-PUFA in plasma phospholipids were affected by *FADS2* genotype distribution regardless of the polyphenol intake. Subjects carrying a variant allele within the rs174616 and rs174576 were with a salient decrease in LA upon 4-weeks duration of the study in comparison with the major homozygote counterparts (Appendix A (*p* = 0.021 and 0.048, respectively)).

We further examined whether the addition of a variant allele within the three *FADS2* polymorphisms modulates anthropometric parameters (Appendix A) and the LC-PUFA (Table 1) and whether the relationship is further affected by the polyphenol interventional treatments. In our study, the addition of a variant allele in either of the *FADS2* variants was not associated with changes in BMI or weight gain; however, the addition of variant alleles was significantly associated with lower LA levels only in subjects drinking AJ. Namely, the addition of a variant allele within the rs174593, rs174616, and rs174576 tended to be linked with −2.12, −2.16, and −1.98 greater decrease in the % of LA in plasma phospholipid pool, respectively, upon controlling for gender and baseline *n* − 6/*n* − 3 intake. The link was also present for the additive effect of rs174576 in subjects drinking MJ, and no effect was observed in subjects drinking PLB (Table 1). A similar trend, but in the opposite direction, was observed for the desaturase-6 predicted activity.

Only in subjects drinking AJ, the addition of the rs174593 was associated with a sustained increase in AA (multivariable-adjusted β = 1.14, *p* = 0.049), reflecting a higher increase in the fatty acid in variant allele carriers consuming polyphenol-rich AJ for the 4-weeks (Table 1).

The addition of rs174616 variant allele in subjects drinking MJ tended to be inversely linked with changes in EPA and DHA levels in plasma phospholipids (multivariable-adjusted β (*p*)= −0.26 (0.041), and −0.70 (0.054), respectively. A similar trend was observed for the desaturase-5 predicted activity (Table 1).

Although without statistical significance, our results demonstrated a trend in dose-dependent changes in BMI and body weight, upon treatments with varying polyphenol content (Figure 1). In line with the demonstrated relationships between *FADS2* make-up in the three genetic variants and LC-PUFAs in plasma phospholipids, we further examined whether the changes in the fatty acids affect changes in the obesity parameters. Our study results demonstrated an inverse relationship between changes in EPA and both BMI and body weight, upon controlling for the baseline *n* − 6/*n* − 3 intake and gender (Table 2). The tendency of the inverse link remained within the subjects drinking MJ upon 4-week study intervention. Further on, when the interaction term was included, regardless of study treatment the inverse relationship tended to be significant for different levels of rs174576, with variant allele carriers having a lower decline in BMI upon EPA increase (*p* for interaction = 0.063, Figure 2a,b). Again, only in subjects drinking AJ, the rs174576 gene-effects tended to affect the link between changes in EPA with BMI and weight gain (*p* = 0.067 for both).

## 4. Discussion

Our study examined a link between habitual polyphenol intake, levels of LC-PUFA, and obesity parameters in adult subjects, and depending on the variant allele presence within the three *FADS2* SNPs (rs174593, rs174616, and rs174576). The main findings of our study demonstrated an inverse relationship between changes in EPA in plasma phospholipids and changes in BMI and weight gain in the subjects drinking polyphenol-rich Aronia juice, which tended to be modified by the presence of the rs174576 variant allele. The results imply that daily polyphenol intake for weight management strategies might depend on the background fat metabolism, and further controlled studies should examine the relationship in larger cohorts. 

To the best of our knowledge, we are the first to report a relationship between *FADS2* variants and polyphenol intake on cardiometabolic traits. Previous nutrigenetic observational study with apparently healthy subjects demonstrated associations between variant alleles in several SNPs in the gene coding for paraoxonase 1, an enzyme involved in HDL-mediated cholesterol clearance, and increased HDL-c levels under high habitual polyphenol and anthocyanin intakes [27]. The influence of polymorphisms in *COMT* (catechol-O-methyltransferase) gene encoding phase II methylation enzyme was observed in the acute effect of green tea polyphenols on arterial stiffness [36]. Moreover, another randomized controlled study showed the interaction between 4-week polyphenol-rich apple juice intervention and genetic variants within the gene encoding for the pro-inflammatory cytokine interleukin-6 on body fat content in obese men [25]. Still, the latter study failed to observe gene-intervention interactions for SNPs within a few genes involved in lipid metabolism and other anthropometric and biochemical parameters examined.

A recent review paper summarized the effects of dietary components including polyphenols, on the transcriptional levels and activity of the desaturase and elongase enzymes, as well as the plasma and tissue LC-PUFA levels [37]. Overall, the current knowledge mostly relies on the experimental data linking intake of polyphenols with transcriptional mechanisms affecting desaturase activity [37]. A study in postmenopausal women demonstrated no differences in EPA, DHA, or predicted desaturase-6 activity in total plasma lipids, upon 12-week intake of anthocyanin-rich elderberry extract [37]. In our study, in subjects consuming AJ with substantial polyphenol content, the addition of a variant allele in either of the *FADS2* variants (rs174593, rs174616, and rs174576) was significantly associated with a lower decrease in plasma phospholipid LA, and rs174593 with a higher increase in AA. The results were followed by respective changes in predicted desaturase-6 activity, indicating the functional capacity of dietary polyphenols in modulating LC-PUFA metabolism. Furthermore, the presence of rs174616 variant allele was associated with lower EPA and DHA levels in the plasma phospholipids, only upon intake of Aronia juice with moderate levels of polyphenols, yet the levels corresponding to doses easily achievable by consuming 100 g of different fruits like blueberries, strawberries, grapes, apples or oranges [38]. Estimated average daily intakes of polyphenols in Europe are generally high, with values ranging from around 1700 mg/day in Denmark and 1193 mg/day in France to about 600 mg/day in Greece or Italy, mainly achieved through consumption of fruits and plant-food derived beverages [19,39]. Although polyphenol concentrations in AJ were substantially higher, the inverse association between EPA and DHA levels in subjects carrying rs174616 variant allele was only observed upon treatment with MJ. It is possible that the lower polyphenol dose was sufficient to achieve this effect, after which, with higher doses, the response plateaued and started to decrease. In support of this speculation, the non-linear dose-response relation was previously observed with different polyphenol-rich beverages and endothelial vasodilatory function [40,41,42]. The non-linear association was also reported between polyphenol exposure and some obesity-related comorbidities such as type-2 diabetes [18]. In line with that, daily consumption of AJ juice containing 1177.11 mg of polyphenols per portion, decreased body weight by a median of 0.25 kg per 4-weeks in our study, falling beyond clinical relevance, given the median of 92.25 kg of the sample. Interestingly, although with a narrower range, the median weight loss in subjects drinking MJ was 0.55 kg, which was superior in comparison with AJ. Our study sample included subjects who were overweight, out of which 41.9% of the subjects were obese, but otherwise healthy people, which might have additionally constrained the clinical relevance of our findings. Also, we evaluated the health effects of dietary polyphenols in the context of the common background diet, and study participants received no dietary instructions nor recommendations for weight loss, again potentially contributing to the lack of clinical effect. Hence, our findings should be replicated with a controlled study in larger cohorts of subjects who are obese. The personalized nutritional advice for polyphenols dosing remains a challenge, and we propose the dose of Aronia juice with up to 300 mg/100 mL in addition to a prudent background diet rich in fruits and vegetables, as suitable for the design of the interventional-controlled studies to achieve clinically relevant results in terms of weight management, regardless of genetic background.

Previous studies delivered inconclusive results on the effects of polyphenols on weight gain. Large prospective studies have reported that higher intakes of several polyphenol groups are associated with less weight gain over 24 years or less BMI gain over 14 years [43,44]. On the other hand, the role of polyphenols in weight management is less clear from clinical trials. Although recent meta-analyses suggested the favorable effect of blueberries on body weight and resveratrol (a polyphenol found in berries and wine) on both body weight and BMI [45,46], data on other polyphenols and polyphenol-rich food sources are inconstant, probably due to the high heterogeneity in study designs [47]. In our previous work, a 4-week treatment with Aronia juice with the same polyphenol content failed to induce changes in body weight [28]. Similar results were reported in apparently healthy women after consuming 400 mg polyphenols/day in Aronia juice for 12 weeks [16]. By contrast, glucomannan-enriched Aronia juice with around 600 mg/100g polyphenols tended to reduce body weight and significantly reduced BMI in women with abdominal obesity after 4 weeks [15], which might be due to the glucomannan effects [48]. Furthermore, a number of epidemiological studies demonstrated a link between fatty acids in plasma phospholipids and parameters of obesity and metabolic syndrome [4,49]. Cross sectional-studies also reported the inverse association between EPA levels in erythrocyte membranes and BMI [50] and subjects who are obese had lower EPA levels in plasma phospholipids in comparison with their lean counterparts [51]. Our results indicated a trend in the AJ-induced higher decrease in anthropometric parameters and AA, followed by a greater increase in EPA and DHA, however, failing to reach statistical significance. The AJ treatment rich in polyphenols, however, exerted significant effects in lowering AA/DHA ratio, the latter investigated as a prognostic biomarker in already established CVD but with net inconclusive findings [52]. It is important to note, that a well-known interindividual variability in the biological response to the polyphenol intake [25], also demonstrated in our results in wide ranges of treatment responses (Figure 1), might have blurred significant effects, and must be taken into account with personalization of dietary advice. Finally, in this study, we showed the relationship between changes in EPA and changes in BMI and total body weight. Practically, upon 4-weeks of MJ, a 1% increase in EPA, which appears as readily achievable with substantial polyphenol doses (Figure 1), would be associated with 0.73 kg/m^2^ lower BMI and weight loss of 2.17 kg. Furthermore, only in subjects drinking AJ, the inverse link between changes in plasma phospholipid EPA levels and BMI and weight gain was affected by the presence of the alternative allele in *FADS2* rs174576 variant. Thus, these findings suggest that the potential role of polyphenols in weight management might be influenced by an individual’s genetic background regarding fat metabolism, in addition to gender and background fat intake. 

One of the possible mechanisms underlying polyphenol effects on body weight might be their ability to inhibit intestinal absorption of fats and carbohydrates, which reduces energy uptake [47,53]. Polyphenols’ capacity to down-regulate various pro-inflammatory adipocytokines also seems to play an important role in their proposed weight control effects [54,55]. Furthermore, a mechanistic insight behind the hypolipidemic effects of Aronia berry has been reported based on its nutrigenomic effect via genes for cholesterol synthesis, uptake and transport, and genes for lipoprotein metabolism [51]. The mechanistic aspect underpinning herein observed pronounced effects on LC-PUFA associations in the overweight and obese subjects consuming polyphenol-rich Aronia juice might similarly include nutrigenomic effects of dietary polyphenols through regulation of PPAR-α activity [37]. Most of the available mechanistic evidence comes from cell and animal studies with only a few specific polyphenols, and further work is needed to elucidate the underlying mechanisms.

Previous studies revealed interactions between fatty acid metabolism and micronutrient status [37,56,57]. Herein, we present novel findings on the interactions between dietary bioactives and fatty acid metabolism. We propose that the inverse effects of circulating EPA on changes in body weight are promoted in the presence of substantial dietary polyphenol doses, and in subjects who are rs174576 reference allele carriers. Circulating EPA is with anti-inflammatory potential, achieved through the net synthesis of pro-resolving eicosanoids [58], and increasing its plasma levels remains one of the optimal strategies for achieving beneficial effects. Our subjects were with substantially high AA/EPA levels (mean ± SD = 41.08 ± 25.33), suggesting salient levels of chronic low-grade inflammation in those subjects, and a greater risk of cardiovascular events [52,59,60]. We however failed to demonstrate the relationship between AA/EPA with obesity parameters, which might be interpreted in light of the fact that AA/EPA has commonly been linked with cardiovascular parameters in subjects with already established disease, such as myocardial infarction survivors [52] or subjects with NAFLD [60], contrasting our sample of apparently healthy subjects. Furthermore, a recent survey reported *n* − 3 status in different blood fractions as weight percentages in erythrocyte equivalents, from versatile worldwide geographical locations, indicating the optimal range of blood EPA and DHA levels mostly in areas not yet fully transitioned to the Westernized dietary habits, such as indigenous populations [58]. Our study sample was with a high *n* − 6/*n* − 3 dietary ratio (median = 17) indicative of adherence to more Westernized dietary patterns, with possible pro-inflammatory outcomes. Our study group across interventional treatments was balanced for background *n* − 6/*n* − 3 intake ratio, and thus we propose that potentiated beneficial effects of EPA on anthropometric parameters in the AJ and MJ groups are due to substantial polyphenol intake. Our results might be of relevance for dietitians and clinical nutritionists practicing a personalized approach in nutritional counseling and preparation of anti-inflammatory dietary plans and based on the incorporation of polyphenol-rich Aronia juice or other food-based polyphenol sources, in precision nutrition plans for the prevention of weight gain. Ideally, dietary counseling might include genotyping of the rs174576, as our previous study has indicated its potential in predicting AA metabolism [28] and the current study paved the way for its potential in modulating the effects of dietary polyphenols. In addition, the presence of a variant allele in either of the *FADS2* variants (rs174593, rs174616, and rs174576) inversely affected LA, and in rs174593 directly affected AA in plasma phospholipids, only in subjects consuming AJ, additionally encouraging *FADS2* genotyping with promotion of polyphenol intake.

Several issues related to our study results warrant attention. Firstly, the limitations of our study are reflected in retrospective analyses resulting in inadequately powered design and failing to reach clinical relevance of the Aronia juice interventional effects. However, this study was a secondary analysis and formed part of a larger study, which was adequately powered for the nutritional and nutrigenetic analyses [28,29]. Secondly, the dietary intake of the 62 subjects included in the current analyses virtually resembled the intake of the initial sample (*p* > 0.05 for independent *t*-test/Mann–Whitney test for each category of the intake). However, the confounding effects of the background diet might not be excluded, and temporal changes in dietary habits fell beyond the scope of this study. At baseline, the dietary intake of the study subjects was balanced across the interventional treatments, and we surmise it did not change over the 4 weeks during the study course. Thirdly, our study included two interventional treatments with varying polyphenol content, however, we failed to demonstrate a dose-dependent effect. We surmise that the dietary effect of polyphenols reaches a plateau with a certain dose, potentially underpinning the fact that in our study interventional effects of AJ did not correspond to the MJ. Thus, the use of the 2 doses might be considered as a disadvantage within our study design, as it might have blurred potentially significant associations. We also demonstrated the relationship between clupanodonic fatty acid and variant allele presence in rs174616 and rs174576 in subjects drinking a polyphenol-lacking placebo beverage. This result is beyond the interpretation of the current findings. We speculate it might be due to chance, as well as possible interaction of the non-polyphenolic nutritional components from the placebo beverage with fatty acid metabolism, which are otherwise outweighed by the presence of dietary polyphenols. Finally, although previous studies supported the utilization of *FADS* nutrigenetic biomarkers in tailoring advice on essential fatty acid intake [61], to the last of our knowledge we are the first to report the interaction of the *FADS2* with dietary components other than fat.

## 5. Conclusions

We found an inverse relationship between changes in EPA and parameters of obesity upon the 4-week interventional study; and based on our findings, increasing EPA intake should be considered as a suitable approach for reducing weight gain. Further on, our results provide insight into the possible genetically-regulated response to polyphenol intake mediated by fatty acid desaturation. Our previous precision nutrition study demonstrated the capacity of the rs174576 as a potential nutrigenetic biomarker for fat intake recommendations in subjects from the general population [28]. In the current study, the same variant allele appeared to modify the relationship between changes in EPA and weight gain in overweight and obese subjects drinking Aronia juice with high polyphenol content, further putting in focus the utilization of the rs174576 *FADS2* variant place in precision nutrition, particularly for the weight management strategies. Further larger-cohort controlled interventional studies are urged to confirm the findings.

## Figures and Tables

**Figure 1 nutrients-13-00296-f001:**
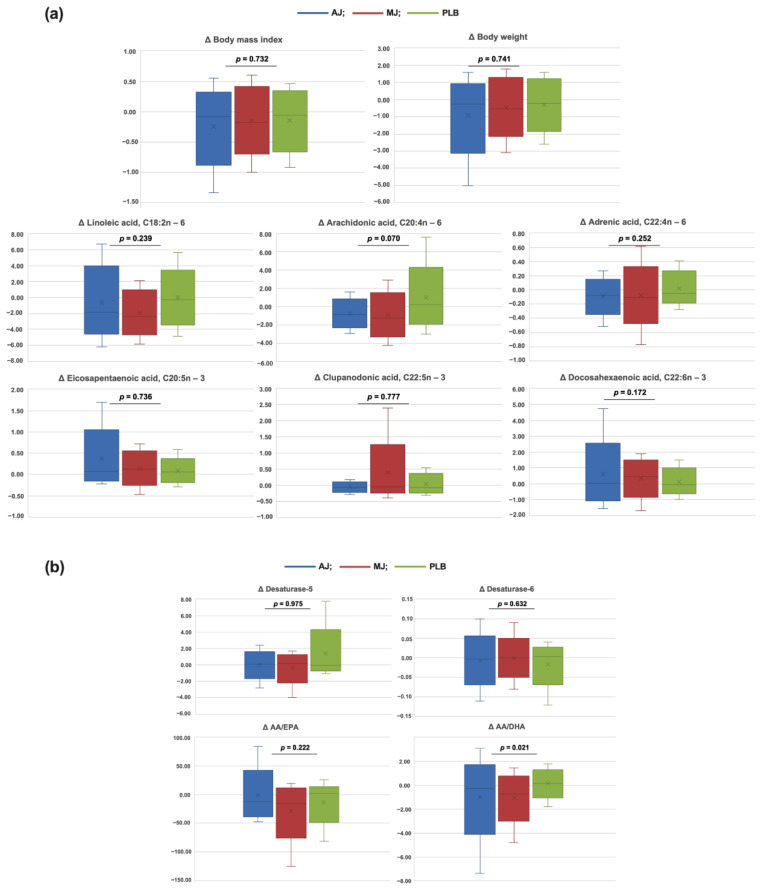
(**a**) Changes in BMI, total body weight, individual long-chain plasma phospholipid fatty acids (**b**) predicted desaturase 5 and 6 activities, AA/EPA and AA/DHA ratios upon 4-week habitual intake of either Aronia juice with 1177.11 mg of polyphenols (gallic acid equivalents, GAE) per 100 mL (AJ), Aronia juice with 294.28 mg of polyphenols (GAE) per 100 mL (MJ), or polyphenol-lacking placebo beverage (PLB). Data are graphed as median [min, max]. Variance in the distribution of the parameters was evaluated by the general linear model and Kruskal-Wallis analyses, for parametric and non-parametric approaches, respectively; testing differences between parameter distributions across the study treatments with varying polyphenol content.

**Figure 2 nutrients-13-00296-f002:**
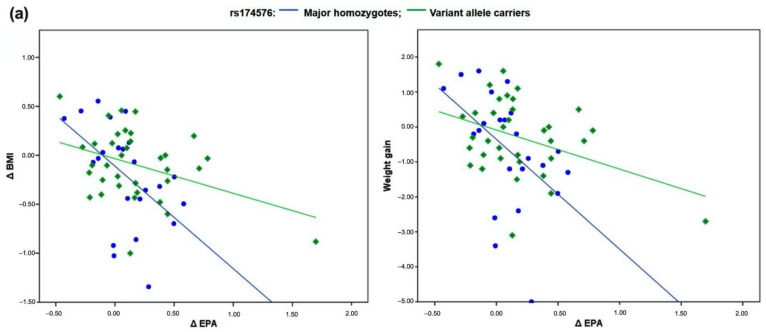
Dietary polyphenol-*FADS2* (rs174576) interaction affecting the relationship between changes in EPA and BMI in overweight subjects (BMI ≥ 25) upon 4-weeks. (**a**) The associations between changes in EPA levels in plasma phospholipids with changes in BMI and weight gain, depending on the variant allele presence in *FADS2* rs174576 variant place; (**b**) The associations between tertiles of changes in levels of EPA in plasma phospholipids with changes in BMI and weight gain, depending on the interventional treatments with varying polyphenol content (AJ, MJ, PLB) and presence of the variant allele in *FADS2* rs174576 variant place. AJ, subjects drinking Aronia juice with 1177.11 mg of polyphenols (gallic acid equivalents, GAE) per 100 mL; MJ, subjects drinking Aronia juice with 294.28 mg of polyphenols (GAE) per 100 mL; PLB, subjects drinking a polyphenol-lacking placebo beverage.

**Table 1 nutrients-13-00296-t001:** Associations between *FADS2* variants (rs174593, rs174616, rs174576) and changes in long-chain polyunsaturated fatty acids in plasma phospholipids in subjects who are overweight (BMI ≥ 25), across the study intervention groups with varying polyphenol intake.

	rs174593			rs174616				rs174576		
		ΔR^2^	B ± S.E.	*p*	ΔR^2^	B ± S.E.	*p*	ΔR^2^	B ± S.E.	*p*
**Δ Linoleic acid, C18:3*n* − 6**										
M1	**TS**	0.001	−0.15 ± 0.57	0.789	0.067	−1.05 ± 0.53	0.051	0.029	−0.72 ± 0.56	0.204
M2		0.042	−0.11 ± 0.58	0.851	0.034	−0.98 ± 0.53	0.070	0.032	−0.57 ± 0.57	0.320
M1	**AJ**	0.250	−2.85 ± 1.11	**0.018**	0.341	−2.47 ± 0.77	**0.004**	0.239	−2.46 ± 0.98	**0.021**
M2		0.181	−2.12 ± 1.06	0.062	0.175	−2.16 ± 0.77	**0.012**	0.197	−1.98 ± 0.97	0.056
M1	**MJ**	0.058	−0.86 ± 0.86	0.337	0.131	−1.33 ± 0.86	0.140	0.345	−2.12 ± 0.73	**0.010**
M2		0.236	−1.16 ± 0.83	0.183	0.193	−1.33 ± 0.81	0.123	0.074	−1.81 ± 0.78	**0.036**
M1	**PLB**	0.149	1.51 ± 0.93	0.126	0.022	0.58 ± 1.00	0.569	0.092	1.19 ± 0.96	0.237
M2		0.110	1.88 ± 1.01	0.085	0.079	0.85 ± 1.13	0.465	0.096	1.51 ± 1.06	0.178
**Δ Arachidonic acid, C20:4*n* − 6**										
M1	**TS**	0.038	0.62 ± 0.43	0.149	0.014	−0.37 ± 0.42	0.382	0.000	−0.07 ± 0.43	0.875
M2		0.012	0.68 ± 0.44	0.130	0.005	−0.34 ± 0.43	0.422	0.006	−0.02 ± 0.45	0.972
M1	**AJ** **MJ**	0.198	1.10 ± 0.50	**0.038**	0.003	−0.10 ± 0.41	0.812	0.078	0.61 ± 0.47	0.208
M2		0.015	1.14 ± 0.54	**0.049**	0.017	−0.11 ± 0.48	0.818	0.033	0.73 ± 0.53	0.183
M1	**MJ**	0.000	0.06 ± 0.78	0.943	0.184	−1.38 ± 0.73	0.076	0.005	−0.23 ± 0.79	0.773
M2		0.014	−0.04 ± 0.86	0.967	0.016	−1.39 ± 0.77	0.094	0.012	−0.18 ± 0.89	0.844
M1	**PLB**	0.000	0.02 ± 0.98	0.987	0.021	−0.54 ± 0.97	0.582	0.117	−1.29 ± 0.92	0.179
M2		0.151	−0.15 ± 1.04	0.886	0.172	−0.87 ± 1.04	0.417	0.201	−1.69 ± 0.94	0.096
**Δ Adrenic acid, C22:4*n* − 6**										
M1	**TS**	0.070	0.09 ± 0.04	**0.047**	0.015	0.04 ± 0.04	0.360	0.126	0.12 ± 0.04	**0.007**
M2		0.040	0.08 ± 0.04	0.063	0.045	0.03 ± 0.04	0.452	0.024	0.11 ± 0.04	**0.016**
M1	**AJ**	0.007	−0.03 ± 0.07	0.708	0.064	0.06 ± 0.05	0.255	0.037	0.05 ± 0.06	0.393
M2		0.303	−0.04 ± 0.06	0.540	0.237	0.02 ± 0.05	0.706	0.259	0.01 ± 0.06	0.881
M1	**MJ**	0.077	0.13 ± 0.11	0.264	0.025	−0.08 ± 0.12	0.533	0.199	0.21 ± 0.11	0.064
M2		0.120	0.13 ± 0.11	0.263	0.121	−0.08 ± 0.12	0.519	0.055	0.18 ± 0.12	0.135
M1	**PLB**	0.203	0.11 ± 0.06	0.070	0.127	0.09 ± 0.06	0.161	0.117	0.08 ± 0.06	0.179
M2		0.101	0.08 ± 0.06	0.199	0.118	0.05 ± 0.06	0.427	0.125	0.05 ± 0.06	0.444
**Δ** **Eicosapentaenoic acid, C20:5 *n* − 3**										
M1	**TS**	0.008	−0.05 ± 0.07	0.516	0.003	0.03 ± 0.07	0.672	0.001	0.02 ± 0.07	0.784
M2		0.020	−0.05 ± 0.08	0.514	0.018	0.03 ± 0.07	0.734	0.018	0.01 ± 0.08	0.917
M1	**AJ**	0.000	−0.01 ± 0.19	0.968	0.100	0.20 ± 0.13	0.152	0.047	0.16 ± 0.16	0.331
M2		0.125	−0.05 ± 0.20	0.813	0.062	0.14 ± 0.15	0.370	0.086	0.09 ± 0.18	0.641
M1	**MJ**	0.079	−0.14 ± 0.12	0.259	0.234	−0.26 ± 0.12	**0.042**	0.021	−0.07 ± 0.13	0.570
M2		0.102	−0.15 ± 0.13	0.253	0.103	−0.26 ± 0.12	**0.041**	0.140	−0.14 ± 0.13	0.323
M1	**PLB**	0.016	0.04 ± 0.09	0.627	0.028	0.06 ± 0.09	0.523	0.001	0.01 ± 0.09	0.903
M2		0.078	0.01 ± 0.10	0.954	0.068	0.02 ± 0.10	0.866	0.101	−0.03 ± 0.10	0.736
**Δ Clupanodonic acid, C22:5*n* − 3**										
M1	**TS**	0.046	0.13 ± 0.08	0.110	0.007	0.05 ± 0.08	0.541	0.024	0.09 ± 0.08	0.254
M2		0.018	0.12 ± 0.08	0.153	0.027	0.05 ± 0.08	0.540	0.029	0.10 ± 0.08	0.236
M1	**AJ**	0.014	0.03 ± 0.05	0.596	0.030	−0.03 ± 0.04	0.438	0.003	0.01 ± 0.05	0.797
M2		0.025	0.02 ± 0.06	0.726	0.082	−0.06 ± 0.04	0.220	0.029	0.00 ± 0.05	0.953
M1	**MJ**	0.045	0.21 ± 0.24	0.396	0.003	0.05 ± 0.25	0.840	0.011	0.11 ± 0.24	0.673
M2		0.021	0.18 ± 0.26	0.507	0.036	0.05 ± 0.26	0.852	0.050	0.17 ± 0.27	0.546
M1	**PLB**	0.196	0.16 ± 0.08	0.075	0.254	0.18 ± 0.08	0.039	0.228	0.17 ± 0.08	0.053
M2		0.042	0.13 ± 0.09	0.174	0.030	0.16 ± 0.09	0.106	0.035	0.15 ± 0.09	0.133
**Δ** **Docosahexaenoic acid, C22:6*n* − 3**										
M1	**TS**	0.004	0.10 ± 0.21	0.630	0.007	−0.12 ± 0.20	0.548	0.014	0.18 ± 0.21	0.388
M2		0.052	0.09 ± 0.21	0.676	0.058	−0.16 ± 0.20	0.439	0.046	0.12 ± 0.21	0.579
M1	**AJ**	0.103	0.77 ± 0.51	0.146	0.010	0.18 ± 0.40	0.658	0.121	0.74 ± 0.44	0.113
M2		0.020	0.68 ± 0.55	0.233	0.038	0.02 ± 0.46	0.968	0.011	0.66 ± 0.51	0.205
M1	**MJ**	0.001	−0.05 ± 0.39	0.908	0.187	−0.69 ± 0.36	0.073	0.003	0.08 ± 0.39	0.832
M2		0.203	0.01 ± 0.38	0.977	0.208	−0.70 ± 0.33	0.054	0.210	−0.16 ± 0.39	0.688
M1	**PLB**	0.011	−0.10 ± 0.24	0.685	0.001	0.03 ± 0.24	0.897	0.008	−0.08 ± 0.24	0.735
M2		0.009	−0.06 ± 0.28	0.822	0.024	0.10 ± 0.29	0.733	0.010	−0.04 ± 0.28	0.879
**Δ Destaurase-5 predicted activity, AA/DGLA**										
M1	**TS**	0.004	−0.15 ± 0.32	0.646	0.034	−0.43 ± 0.31	0.167	0.077	−0.66 ± 0.31	**0.037**
M2		0.050	−0.12 ± 0.32	0.718	0.044	−0.38 ± 0.31	0.219	0.031	−0.58 ± 0.31	0.073
M1	**AJ**	0.097	−0.78 ± 0.53	0.159	0.001	−0.06 ± 0.42	0.880	0.135	−0.81 ± 0.46	0.092
M2		0.070	−0.62 ± 0.56	0.284	0.124	0.25 ± 0.46	0.585	0.042	−0.62 ± 0.51	0.241
M1	**MJ**	0.002	−0.09 ± 0.49	0.851	0.420	−1.30 ± 0.38	**0.004**	0.050	−0.44 ± 0.48	0.372
M2		0.036	−0.19 ± 0.53	0.729	0.033	−1.31 ± 0.40	**0.005**	0.021	−0.43 ± 0.54	0.441
M1	**PLB**	0.002	−0.15 ± 0.76	0.850	0.023	−0.45 ± 0.75	0.562	0.134	−1.08 ± 0.71	0.148
M2		0.153	−0.24 ± 0.81	0.771	0.165	−0.63 ± 0.81	0.450	0.189	−1.33 ± 0.73	0.091
**Δ Destaurase-6 predicted activity, DGLA/LA**										
M1	**TS**	0.041	0.01 ± 0.01	0.130	0.075	0.02 ± 0.01	**0.039**	0.139	0.02 ± 0.01	**0.004**
M2		0.061	0.01 ± 0.01	0.128	0.051	0.02 ± 0.01	0.054	0.035	0.02 ± 0.01	**0.010**
M1	**AJ**	0.232	0.04 ± 0.02	**0.023**	0.095	0.02 ± 0.01	0.163	0.332	0.04 ± 0.01	**0.005**
M2		0.118	0.03 ± 0.02	0.052	0.116	0.01 ± 0.01	0.551	0.054	0.03 ± 0.01	**0.029**
M1	**MJ**	0.018	0.01 ± 0.02	0.600	0.157	0.03 ± 0.02	0.103	0.103	0.02 ± 0.02	0.194
M2		0.030	0.01 ± 0.02	0.525	0.020	0.03 ± 0.02	0.123	0.010	0.02 ± 0.02	0.243
M1	**PLB**	0.002	0.00 ± 0.02	0.882	0.018	0.01 ± 0.02	0.604	0.067	0.02 ± 0.01	0.316
M2		0.071	0.00 ± 0.02	0.878	0.075	0.01 ± 0.02	0.577	0.081	0.02 ± 0.02	0.298
**Δ** **(Arachidonic acid/Eicosapentaenoic acid)**										
M1	**TS**	0.003	2.54 ± 6.19	0.684	0.003	2.38 ± 5.98	0.692	0.000	0.54 ± 6.18	0.931
M2		0.046	4.47 ± 6.28	0.480	0.042	3.14 ± 5.99	0.602	0.042	1.93 ± 6.29	0.760
M1	**AJ**	0.015	−6.89 ± 12.33	0.582	0.026	−6.69 ± 9.09	0.470	0.043	−10.12 ± 10.71	0.356
M2		0.017	−4.68 ± 13.47	0.732	0.014	−5.53 ± 10.60	0.608	0.012	−9.14 ± 12.19	0.463
M1	**MJ**	0.000	−0.09 ± 13.91	0.995	0.060	14.12 ± 13.94	0.326	0.000	0.19 ± 14.11	0.989
M2		0.144	2.71 ± 14.19	0.851	0.146	14.58 ± 13.70	0.305	0.148	4.76 ± 14.71	0.751
M1	**PLB**	0.045	7.06 ± 8.41	0.414	0.000	0.72 ± 8.60	0.934	0.027	5.44 ± 8.49	0.531
M2		0.055	10.20 ± 9.45	0.300	0.026	3.17 ± 10.02	0.756	0.047	8.47 ± 9.63	0.395
**Δ (Arachidonic acid/Docosahexaenoic acid)**										
M1	**TS**	0.001	−0.09 ± 0.35	0.789	0.001	0.09 ± 0.34	0.792	0.015	−0.31 ± 0.34	0.369
M2		0.047	−0.06 ± 0.35	0.872	0.049	0.14 ± 0.34	0.669	0.039	−0.21 ± 0.35	0.551
M1	**AJ**	0.107	−1.32 ± 0.85	0.138	0.008	−0.27 ± 0.67	0.692	0.151	−1.38 ± 0.73	0.073
M2		0.017	−1.16 ± 0.93	0.230	0.041	−0.05 ± 0.77	0.954	0.018	−1.34 ± 0.83	0.124
M1	**MJ**	0.001	−0.08 ± 0.59	0.898	0.028	0.41 ± 0.60	0.508	0.000	0.05 ± 0.60	0.940
M2		0.141	−0.21 ± 0.60	0.729	0.135	0.41 ± 0.60	0.503	0.155	0.36 ± 0.62	0.569
M1	**PLB**	0.024	0.20 ± 0.32	0.550	0.004	−0.08 ± 0.33	0.819	0.017	−0.17 ± 0.33	0.616
M2		0.174	0.11 ± 0.34	0.746	0.216	−0.23 ± 0.34	0.508	0.226	−0.31 ± 0.33	0.362

M1, Crude regression model; M2, further adjusted for gender and habitual *n* − 6/*n* − 3 intake. TS, Total study sample; AJ, subjects drinking Aronia juice with 1177.11 mg of polyphenols (gallic acid equivalents, GAE) per 100 mL; MJ, subjects drinking Aronia juice with 294.28 mg of polyphenols (GAE) per 100 mL; PLB, subjects drinking a polyphenol-lacking placebo beverage. p, probability values associated with unstandardized coefficients (B) upon multivariate-adjusted, hierarchical linear regression. Bolded text denotes significance (*p* < 0.05). AA, arachidonic acid; DGLA, dihomo-γ-linolenic acid; LA, linoleic acid.

**Table 2 nutrients-13-00296-t002:** Relationship between changes in long-chain polyunsaturated fatty acids in plasma phospholipids with BMI and total body weight in subjects who are overweight (BMI ≥ 25), across the study intervention groups with varying polyphenol intake.

		Δ BMI			Δ Body Weight		
		ΔR^2^	B ± S.E.	*p*	ΔR^2^	B ± S.E.	*p*
	**Δ Linoleic acid, C18:2*n* − 6**						
M1	**TS**	0.001	0.01 ± 0.02	0.825	0.002	0.02 ± 0.07	0.729
M2		0.040	0.00 ± 0.02	0.929	0.045	0.00 ± 0.07	0.985
M1	**AJ**	0.000	0.00 ± 0.04	0.998	0.002	0.02 ± 0.12	0.863
M2		0.229	−0.04 ± 0.04	0.291	0.232	−0.12 ± 0.13	0.341
M1	**MJ**	0.000	0.00 ± 0.05	0.937	0.000	0.00 ± 0.15	0.994
M2		0.150	−0.02 ± 0.06	0.674	0.150	−0.09 ± 0.16	0.587
M1	**PLB**	0.007	0.01 ± 0.04	0.755	0.007	0.04 ± 0.11	0.754
M2		0.027	0.01 ± 0.04	0.888	0.026	0.02 ± 0.12	0.884
	**Δ Arachidonic acid, C20:4*n* − 6**						
M1	**TS**	0.014	−0.02 ± 0.03	0.387	0.007	−0.05 ± 0.09	0.542
M2		0.045	−0.03 ± 0.03	0.323	0.050	−0.07 ± 0.09	0.459
M1	**AJ**	0.042	−0.07 ± 0.08	0.358	0.036	−0.23 ± 0.26	0.397
M2		0.177	−0.07 ± 0.08	0.340	0.190	−0.22 ± 0.25	0.390
M1	**MJ**	0.106	−0.08 ± 0.06	0.187	0.091	−0.20 ± 0.16	0.222
M2		0.142	−0.08 ± 0.05	0.174	0.138	−0.21 ± 0.16	0.204
M1	**PLB**	0.002	0.01 ± 0.04	0.859	0.006	0.03 ± 0.11	0.771
M2		0.046	0.02 ± 0.04	0.652	0.049	0.08 ± 0.13	0.574
	**Adrenic acid, C22:4*n* − 6**						
M1	**TS**	0.008	−0.18 ± 0.27	0.521	0.005	−0.46 ± 0.85	0.597
M2		0.035	−0.10 ± 0.28	0.735	0.042	−0.18 ± 0.87	0.834
M1	**AJ**	0.004	0.19 ± 0.65	0.775	0.003	0.52 ± 2.17	0.812
M2		0.242	0.91 ± 0.71	0.218	0.245	2.73 ± 2.37	0.263
M1	**MJ**	0.129	−0.56 ± 0.36	0.143	0.121	−1.56 ± 1.05	0.156
M2		0.070	−0.41 ± 0.39	0.321	0.067	−1.14 ± 1.15	0.340
M1	**PLB**	0.048	0.51 ± 0.58	0.397	0.044	1.46 ± 1.76	0.421
M2		0.038	0.60 ± 0.69	0.398	0.034	1.70 ± 2.09	0.431
	**Δ** **Eicosapentaenoic acid, C20:5*n* − 3**						
M1	**TS**	0.147	−0.47 ± 0.15	**0.003**	0.142	−1.45 ± 0.48	**0.004**
M2		0.023	−0.45 ± 0.16	**0.006**	0.028	−1.36 ± 0.49	**0.007**
M1	**AJ**	0.122	−0.36 ± 0.22	0.112	0.103	−1.11 ± 0.73	0.145
M2		0.116	−0.27 ± 0.23	0.252	0.137	−0.81 ± 0.76	0.303
M1	**MJ**	0.264	−0.73 ± 0.30	**0.029**	0.280	−2.17 ± 0.87	**0.024**
M2		0.051	−0.63 ± 0.33	0.078	0.043	−1.89 ± 0.95	0.066
M1	**PLB**	0.067	−0.42 ± 0.41	0.316	0.069	−1.30 ± 1.23	0.307
M2		0.028	−0.43 ± 0.45	0.358	0.029	−1.34 ± 1.37	0.344
	**Δ Clupanodonic acid, C22:5*n* − 3**						
M1	**TS**	0.004	−0.08 ± 0.15	0.625	0.003	−0.20 ± 0.48	0.682
M2		0.043	−0.10 ± 0.16	0.544	0.049	−0.27 ± 0.49	0.583
M1	**AJ**	0.001	−0.09 ± 0.82	0.912	0.000	−0.24 ± 2.74	0.933
M2		0.180	0.19 ± 0.80	0.813	0.195	0.73 ± 2.64	0.785
M1	**MJ**	0.032	−0.13 ± 0.18	0.477	0.028	−0.36 ± 0.53	0.506
M2		0.147	−0.15 ± 0.18	0.421	0.142	−0.43 ± 0.54	0.433
M1	**PLB**	0.011	0.16 ± 0.40	0.686	0.007	0.40 ± 1.21	0.748
M2		0.031	0.17 ± 0.45	0.715	0.030	0.38 ± 1.36	0.785
	**Δ** **Docosahexaenoic acid, C22:6*n* − 3**						
M1	**TS**	0.008	−0.04 ± 0.06	0.512	0.004	−0.09 ± 0.18	0.624
M2		0.035	−0.02 ± 0.06	0.749	0.043	−0.02 ± 0.18	0.901
M1	**AJ**	0.010	0.04 ± 0.08	0.664	0.013	0.14 ± 0.27	0.609
M2		0.206	0.07 ± 0.08	0.365	0.226	0.27 ± 0.26	0.306
M1	**MJ**	0.133	−0.17 ± 0.11	0.137	0.117	−0.46 ± 0.32	0.165
M2		0.064	−0.13 ± 0.13	0.328	0.060	−0.32 ± 0.37	0.397
M1	**PLB**	0.025	−0.09 ± 0.15	0.541	0.034	−0.33 ± 0.45	0.476
M2		0.034	−0.10 ± 0.16	0.549	0.033	−0.34 ± 0.48	0.489
	**Δ Desaturase-5 predicted activity, AA/DGLA**						
M1	**TS**	0.003	0.02 ± 0.04	0.677	0.003	0.05 ± 0.12	0.697
M2		0.038	0.00 ± 0.04	0.934	0.044	0.00 ± 0.12	0.974
M1	**AJ**	0.007	−0.03 ± 0.08	0.704	0.007	−0.10 ± 0.26	0.705
M2		0.228	−0.09 ± 0.08	0.261	0.245	−0.30 ± 0.25	0.246
M1	**MJ**	0.134	0.14 ± 0.09	0.135	0.142	0.41 ± 0.25	0.123
M2		0.142	0.14 ± 0.09	0.125	0.129	0.41 ± 0.25	0.123
M1	**PLB**	0.003	−0.01 ± 0.05	0.844	0.004	−0.04 ± 0.15	0.813
M2		0.030	0.00 ± 0.06	0.945	0.027	0.00 ± 0.17	0.981
	**Δ Desaturase-6 predicted activity, DGLA/LA**						
M1	**TS**	0.012	−1.15 ± 1.40	0.418	0.008	−2.87 ± 4.40	0.516
M2		0.032	−0.66 ± 1.45	0.649	0.040	−1.19 ± 4.53	0.794
M1	**AJ**	0.000	0.06 ± 2.44	0.981	0.000	−0.05 ± 8.12	0.995
M2		0.223	2.58 ± 2.53	0.321	0.237	8.51 ± 8.34	0.321
M1	**MJ**	0.224	−4.99 ± 2.32	**0.047**	0.216	−14.19 ± 6.76	0.052
M2		0.134	−4.98 ± 2.27	**0.046**	0.122	−13.99 ± 6.71	0.056
M1	**PLB**	0.027	1.59 ± 2.49	0.532	0.041	6.02 ± 7.48	0.434
M2		0.020	1.22 ± 2.75	0.664	0.016	5.00 ± 8.26	0.556
**Δ** **(Arachidonic acid/ Eicosapentaenoic acid)**							
M1	**TS**	0.042	0.00 ± 0.00	0.127	0.038	0.01 ± 0.01	0.146
M2		0.035	0.00 ± 0.00	0.154	0.042	0.01 ± 0.01	0.172
M1	**AJ**	0.030	0.00 ± 0.00	0.441	0.026	0.01 ± 0.01	0.470
M2		0.161	0.00 ± 0.00	0.588	0.176	0.01 ± 0.01	0.638
M1	**MJ**	0.057	0.00 ± 0.00	0.339	0.056	0.01 ± 0.01	0.346
M2		0.095	0.00 ± 0.00	0.641	0.091	0.01 ± 0.01	0.621
M1	**PLB**	0.061	0.00 ± 0.00	0.339	0.056	0.01 ± 0.01	0.361
M2		0.030	0.00 ± 0.00	0.374	0.030	0.01 ± 0.01	0.395
**Δ (Arachidonic acid/ Docosahexaenoic acid)**							
M1	**TS**	0.007	−0.02 ± 0.04	0.537	0.007	−0.07 ± 0.11	0.525
M2		0.051	−0.04 ± 0.04	0.334	0.058	−0.11 ± 0.11	0.313
M1	**AJ**	0.037	−0.04 ± 0.05	0.390	0.041	−0.15 ± 0.16	0.365
M2		0.220	−0.06 ± 0.05	0.183	0.241	−0.22 ± 0.15	0.152
M1	**MJ**	0.000	−0.01 ± 0.08	0.937	0.001	−0.03 ± 0.22	0.912
M2		0.154	−0.04 ± 0.08	0.614	0.151	−0.14 ± 0.24	0.568
M1	**PLB**	0.013	0.05 ± 0.11	0.667	0.025	0.21 ± 0.34	0.547
M2		0.063	0.10 ± 0.13	0.450	0.071	0.37 ± 0.39	0.353

M1, Crude regression model; M2, further adjusted for gender and habitual *n* − 6/*n* − 3 intake. TS, Total study sample; AJ, subjects drinking Aronia juice with 1177.11 mg of polyphenols per 100 mL; MJ, subjects drinking Aronia juice with 294.28 mg of polyphenols per 100 mL; PLB, subjects drinking a polyphenol-lacking placebo beverage. p, probability values associated with unstandardized coefficients (B) upon multivariate-adjusted, hierarchical linear regression. Bolded text denotes significance (*p* < 0.05). AA, arachidonic acid; DGLA, dihomo-γ-linolenic acid; LA, linoleic acid.

## Data Availability

The data presented in this study are available on request from the corresponding author. The data are not publicly available due to or ethical reason.

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
