# Peer review of "Is There a FADS2-Modulated Link between Long-Chain Polyunsaturated Fatty Acids in Plasma Phospholipids and Polyphenol Intake in Adult Subjects Who Are Overweight?"

_nutrients, 2021, doi:10.3390/nu13020296_

Round 1
Reviewer 1 Report
The submitted manuscript starts with the broad goal of linking the levels of PUFA and polyphenol supplementation to obesity. Relative to that goal, as measured by changes in BMI or weight, the addition of polyphenols had no statistically significant benefits. Even when dividing the entire group into various SNPs of the FADS2 gene, the results were still non-significant. With further subdivision in Table 3, it only with an increase in EPA is there a statistically significant reduction in the change of body weight or BMI upon supplementation using the entire group as measured using both a crude regression model (M1) as well as a more refined adjusted regression model (M2) taking into account gender and dietary intake of n-6 and n-3 fatty acids. However, further analysis of the inverse relationship between EPA and weight change indicated it is only with the moderate level of polyphenol supplementation that had an inverse relationship between EPA and changes in BMI or weight. However, this inverse relationship between EPA levels and weight loss in the moderate polyphenol group was only statistically significant with the crude regression model (M1) but no longer statistically significant using the more refined regression model (M2). It should also be noted that even in the last subdivision of the data, there was no relationship between the AA or DHA levels and weight or BMI. This result is odd since all three fatty acids (AA, EPA, and DHA) are dependent on delta-5 and delta-6 desaturase activity for their synthesis from shorter-chain omega-6 (linoleic acid) and omega-3 fatty acids (alpha-linolenic acid).
At this point, I would generally recommend rejecting the paper. However, I believe some potentially meaningful information can be in the raw clinical data is reinterpreted by the authors to give statistical significance.
This manuscript is a continuation of their previous research on polyphenol supplementation on cardiovascular risk factors (1). In that study, the highest level of polyphenol supplement had a non-significant trend in decreasing BMI. In contrast, the moderate polyphenol supplementation at no effect on BMI, suggesting a potential dose-response effect. A potential dose-response effect at the highest level of polyphenol supplementation is also suggested because they found a statistically significant impact on reducing diastolic blood pressure and oxidized LDL levels. Finally, in this study, both doses of the polyphenol supplement reduced the AA/EPA ratio in the subjects with statistical significance.
With that background, let me return to the submitted manuscript. Obesity and its co-morbidities, such as diabetes, hypertension, and cardiovascular disease, are strongly associated with insulin resistance. Insulin resistance is driven by increased levels of pro-inflammatory cytokines and, in particular, TNFα. It has been demonstrated in both normal-weight subjects and obese subjects that supplementation with omega-3 fatty acids (EPA and DHA) can reduce the AA/EPA ratio with a simultaneous reduction of TNFα and other inflammatory cytokines (2,3). Polyphenols can also reduce cytokines indirectly by activating AMPK. Once AMPK is activated, it leads to NF-kB inhibition, thus reducing cytokine generation (4,5). If cytokines, such as TNFα, are reduced, there should be a reduction in insulin resistance.
Omega-3 fatty acids are also substrates for resolvin formation that also activate the AMPK via FPR2/ALX receptors. Thus, a decrease in the AA/EPA ratio with polyphenol supplementation could increase the likelihood of increased resolvin synthesis. Therefore, this may be a potential mechanistic pathway (although a complex one) between the decreased AA/EPA ratio and weight loss changes mediated by reduced insulin resistance mediated by the polyphenol activation of AMPK.
I would suggest the following items for the resubmission of the manuscript. First, focus on the AA/EPA ratio, not individual fatty acids, for the reanalysis. From their earlier study, the average starting AA/EPA ratio was 35, which suggests significant levels of chronic low-level inflammation in those patients. I suspect the subjects in the submitted study would also have a similar high AA/EPA ratio, so this should be mentioned. Second, I would reanalyze the polyphenol supplement data to see if there is a dose-response effect of the polyphenols on reducing the AA/EPA ratio as they have found in their previous study (1). Third, I would reanalyze the changes in BMI and weight versus differences in the AA/EPA ratio. If these items are done, I believe the existing clinical data will become more statistically significant and pertinent to the subject of weight loss.
If I am correct in my assumptions, then the potential follow-on studies combining omega-3 fatty acid and polyphenol supplementation may have synergistic effects, which would have great relevance to weight reduction and the treatment of cardiovascular disease.
References
- Pokimica B. et al. Nutrients 11: 850 (2019)
- Endres S. et al. N Engl J Med 320:265-71 (1989)
- Tan A. et al. Prostaglandins Leukot Essent Fatty Acids 132:23-29 (2018)
- Chu AJ. Inflamm Allergy Drug Targets 13:34-64 (2014)
- Tian Y. et al. Mol Cell Biochem 422:75-84 (2016)
Author Response
Authors Reply: We would like to thank the Reviewer for providing the insight into additional molecular mechanisms behind proposed interaction between polyphenol intake and fatty acid metabolism. Also, the AA/EPA ratio in phospholipids is an emerging CVD prognostic biomarker, and we acknowledge the importance of reanalyzing the data. To address the issues raised, we have performed below listed analyses, followed by additional comments and discussion along the manuscript. Besides AA/EPA, our analyses also included AA/DHA to additionally account for the competitiveness between omega-3 and omega-6 pathways.
- We included data on the baseline distribution of the AA/EPA and AA/DHA ratios in Table S2. We also additionally commented on the relevance of AA/EPA and AA/DHA ratios as emerging prognostic biomarkers in CVD, as seen in Discussion, L401-403, and we pointed out the importance of AA/EPA as a salient CVD biomarker (which even proved to be superior in comparison with AA/DHA), as evidenced in the most of the up-to-date studies.
- We additionally analyzed the treatment effects on the changes in AA/EPA and AA/DHA ratios, and included those in the Figure 1, now re-structured to Figure 1a and Figure 1b. Our results failed to demonstrate significant treatment effects on AA/EPA, however, we showed significantly stronger effects on AA/DHA decrease with the AJ treatment, containing the highest amounts of polyphenols, and presented those within Figure 1b. Although the capacity of AA/DHA in CVD prognosis might be blurred with certain factors such as statin therapy (PMID: 28125968), the importance of omega-6/omega-3 ratio remains crucially important for CVD primary prevention (PMID: 27843563), further emphasizing the relevance of AJ-polyphenol rich treatment in beneficial health response.
- In line with our initial study design, we further wanted to see whether there was an interaction between polyphenol intake and fatty acid metabolism, and parameters of obesity. The differences in changes in the ratios respective of a variant allele presence within the three SNPs are presented in Table S4 and Table 1 (Previously Table 2).
- Finally, as suggested by the reviewer, we explored whether the change in one unit of either of the ratios (AA/EPA or AA/DHA) was linked with parameters of obesity. The additional results finally complemented Table 2 (Previously table 3), however, we failed to reach statistical significance.
At this point, we would like to tackle the possible causes resulting in a lack of statistical significance in our study for the ratios. We applied robust statistical methods (in comparison with Pokimica et al., where some of the statistical significance results from within treatment-arm analyses, which although supporting the clinical relevance of the results, might appear as biased), in which we analyzed between treatment effects, while in the multivariable-adjusted regression analyses (although suffering from constraint study sample) we controlled for background omega-6/omega-3 intake and gender effects (both known to influence fatty acid metabolism). Secondly, the relevance of AA/EPA has extensively been studied in subjects with already established disease, as well as in a secondary prevention setting L435-439. Our study subjects were apparently healthy, which might have limited the potential of capturing statistical significance for the AA/EPA ratio. The effects of omega-3 in a primary prevention setting remain inconclusive, and we urge for further studies in the field that would examine the role of AA/EPA ratio as well. Although without reaching statistical significance, we incorporated the results on the ratios within the existing Table 1 and Table 2, which would, hopefully, pave the way for the continuing studies in the field. In light with that, we trust our findings on the interrelationship between polyphenol intake, EPA in plasma phospholipids, FADS genetics and obesity, might be a solid background for future study designs.
Reviewer 2 Report
In this work, Glibetic and cols present an interesting study about the possibility of interaction between genetic variability and the effect of functional food on physiological parameters. Actually, the field of study is very interesting, although the low number of subject include in the study, the high number of variables analysed and the complex of the data, makes it difficult to understand the results and conclusions
Mayor issues:
The format and the edition of the manuscript should be revised. Some aspects could be confuses and a lot of result are include in the tables, although most of them are not significant. Significant results could be highlight in the tables.
How could the authors explain that a genetic polymorphism for a desaturase activity affects the concentration of fatty acids in plasma, only when the subjects ingest a certain amount of polyphenols?.
For AJ group they found a relationship between rs174593 allele and araquidonic acid levels in plasma, but not between desaturase activity and arachidonic acid in plasma. However, desaturase activity correlate with rs174576 allele in the same group. Could authors explain this result?.
Authors only found an effect over obesity (less increase of BMI) in MJ group. And in this group the presence of rs174616 allele correlate both with desaturase activity and plasmatic EPA. This result is confuse for mi.
Minor issues:
Line 6: It is not necessary include de number of the participants in the title.
Line 21: Could the authors justifies why they decided to study the aronia juice?
Line 22: It is a little bit confuse; this group consume one aronia juice without polyphenols?
Line 27: What is β?. Please define it
Line 30: The final conclusion at the end of the abstract should be improve
Line 34: Lines 34-35 must be move
Line 101: The Folin-Ciocalteu method estimate the total antioxidant capacity, not only the polyphenols, including other antioxidant molecules such as vitamin E.
Line 107: What time over the day was the juice intake?. Indicate it could be important to avoid Chrono biological differences
Line 119: Have estimated the authors the polyphenols intake of the participants, using, for example the phenol explored platform?
Line 144: Several characters are missing along the manuscript
Line 166: Please, provide the primers used in the RT-PCR
178: Why the waist and hip circumference, and its relation did not include in the analysis?. The waist/hip circumference is a best predictor of cardiovascular risk than the body weight or BMI.
Line 196: Study include only overweight subject, no obese?. If then, please indicate the range of BMI
Line 196: Could authors include ± SEM of age?
Line 199: It is necessary indicated the methodology to determine the intake of PUFA and others FA in the diet
Table 3: Please Body Weight, not Weight
Line 303: Please, define COMT
Appendix Table A1: Please, check the data. Total number subjects = 62. Subjects rs174593 = 35, Subjects rs174616 = 48, Subjects rs174576 = 39; SUM = 122
Appendix Table A1: The total intake of fat (%TE) seem to be very high, corresponding to HFD
Line 685: Please, review the format
Author Response
Comments and Suggestions for Authors
In this work, Glibetic and cols present an interesting study about the possibility of interaction between genetic variability and the effect of functional food on physiological parameters. Actually, the field of study is very interesting, although the low number of subject include in the study, the high number of variables analysed and the complex of the data, makes it difficult to understand the results and conclusions
Authors Reply: Thank you for the notion, we also believe our results tackle important interactions between polyphenol intake and fat metabolism.
Mayor issues:
The format and the edition of the manuscript should be revised. Some aspects could be confuses and a lot of result are include in the tables, although most of them are not significant. Significant results could be highlight in the tables.
Authors Reply: Thank you for the comment, we additionally put efforts to clarify the results at certain point, and to double-check weather all of the statistically significant results have been bolded. We have also moved previous Table 1 to Supplementary materials (now Table S5) to achieve additional clarity and focus on the main results.
How could the authors explain that a genetic polymorphism for a desaturase activity affects the concentration of fatty acids in plasma, only when the subjects ingest a certain amount of polyphenols?.
Authors Reply: We are grateful for the important notion. To the best of our knowledge, we are the first to demonstrate interactive effects of FADS2 gene variants and ingested polyphenols on cardiometabolic traits, including changes in plasma LC-PUFA concentrations. In gene-polyphenol interaction, it is possible that the gene effects manifest depending on the amount of polyphenols, which we have confirmed for the FADS2 variants in the present study. Previous nutrigenetic studies had also reported gene-polyphenol interactions that affected CVD-related traits, such as arterial stiffness (PMID: 22707271), body fat content (PMID: 22038464) and HDL-cholesterol levels (PMID: 27338244). Comparable to our findings, between high and low polyphenol intake, only under the high intake of polyphenols the PON1 genetic variants exerted the cardioprotective effects through association with increased HDL-cholesterol levels (PMID: 27338244). The target FADS2 gene variants have already been associated with LC-PUFA concentrations/estimated desaturase activity, however, their precise effects on expression and activity of the desaturase remain to be determined. Since these polymorphisms are located in the non-coding region of FADS2 gene, their possible direct impact on FADS2 gene transcription is predicted, which could be influenced by polyphenols as these bioactives have been found to affect the FADS2 transcription through regulation of transcription factors such as PPAR-α (PMID: 33186986). With reference to the Reviewer’s comment, this suggested FADS2 gene variant-polyphenol interaction in regulation of FADS2 expression could depend on quantitative polyphenol intake, given that the amount of consumed polyphenols is related to the polyphenol bioavailability and, hence, action as well. Thus, our findings represent a baseline for further studies, which are required to elucidate the precise mechanisms of gene-polyphenol interactions with regard to the role of the investigated FADS2 polymorphisms in regulation of LC-PUFA levels.
For AJ group they found a relationship between rs174593 allele and araquidonic acid levels in plasma, but not between desaturase activity and arachidonic acid in plasma. However, desaturase activity correlate with rs174576 allele in the same group. Could authors explain this result?.
Authors’ reply: We acknowledge the use of estimated parameters for the prediction of the desaturase activities. The desaturase enzymes are shared between omega-6 and omega-3 pathways, and it is known, that in subjects with a low omega-3 intake (as in our study sample), the desaturation pathway is potentiated towards omega-3 longer-chain products. Thus, the estimation of the desaturase activity by the use of the omega-6 metabolic products might underpin herein observed discrepancy.
Authors only found an effect over obesity (less increase of BMI) in MJ group. And in this group the presence of rs174616 allele correlate both with desaturase activity and plasmatic EPA. This result is confuse for mi.
Authors’ reply: One of the drawbacks of the presented study is the use of moderate vs high-polyphenol dose (MJ vs AJ), as previous studies indicate that increasing levels of polyphenols exert dose-dependent effects to a certain extent, after which the effects reach a plateau, which might have underpinned the observed effects of MJ in our study. We discussed the issue within L363-368. In addition, the effect of MJ treatment on BMI was not significant. In MJ-treated subjects the presence of rs174616 alternative allele was associated with decreased delta-5 desaturase activity (AA/DGLA ratio), which pointed to AA decrease relative to DGLA increase. When delta-5 desaturase activity is decreased it is expected that, in parallel with AA decrease, there is a decrease in EPA along with increase in ETA (since delta-5 desaturase drives the conversion of ETA to EPA in n-3 fatty acid pathway), and the latter is consistent with our finding regarding the EPA change in MJ- treated rs174616 alternative allele carriers.
Minor issues:
Line 6: It is not necessary include de number of the participants in the title.
Authors’ reply: We have excluded the number of study participants from the title.
Line 21: Could the authors justifies why they decided to study the aronia juice?
Authors’ reply: Thank you for this comment. We decided to use aronia (chokeberry) juice since it is a rich source of polyphenols that provides a substantial quantity of these phytochemicals in an easily consumed volume of this beverage. To emphasize the important notion, we included in the Abstract of the revised manuscript (Lines 20-21) that the aronia juice is a rich source of polyphenols. We additionally emphasized it in the Introduction. L70-72. Last but not the least, the results present continuation in the research from our group dealing with diverse effects of aronia juice rich in polyphenols.
Line 22: It is a little bit confuse; this group consume one aronia juice without polyphenols?
In this study, for the control treatment we used a previously designed placebo drink (PMID: 28147889) that matched the nutrient content of Aronia juice but lacked polyphenols. The design allowed us to discuss the dietary effects of total phenolic compounds contained in Aronia juice. However, the effects of non-polyphenolic compounds from Aronia juice cannot be ruled out. In addition, in the revised manuscript we clarified that nutritionally matched polyphenol-free placebo drink was used as a control (Lines 22-23).
Line 27: What is β?. Please define it
Authors’ reply: Thank you for the important notion, given that all of the abbreviations should be explained with the first mentioning in the abstract section. The parameter β is a statistical parameter referring to the unstandardized coefficient resulting from a regression analysis. As the nomenclature is commonly used to designate regression coefficients, we believe additionally explaining it might appear redundant, and thus would choose to leave as is.
Line 30: The final conclusion at the end of the abstract should be improve
Authors’ reply: Thank you, we’ve adjusted the conclusion sentence and hopefully in this format it reads more clearly.
Line 101: The Folin-Ciocalteu method estimate the total antioxidant capacity, not only the polyphenols, including other antioxidant molecules such as vitamin E.
Authors’ reply: The Reviewer raised an interesting point. The Folin-Ciocalteu method is regularly used to study the total content of polyphenolic compounds in foods. It is known that other non-phenolics compounds may react with the Folin-Ciocalteu reagent (like some vitamins and their derivatives) which can lead to higher estimated values (PMID: 20583841); however, because the polyphenolic compounds are the most abundant antioxidants in plants, it is generally accepted they have the largest contribution to the values obtained using this assay. Thus, most of the available studies in the literature examining the effects of different polyphenol-rich food products on human health mainly use Folin-Ciocalteu assay to estimate their total polyphenol content. More importantly, in this study, we used a placebo drink that was macro- and micro-nutrient matched to aronia juice but lacked polyphenols to attribute the obtained results to the effect of polyphenols and not some other antioxidant component of the juice. We trust the notion is of substantial relevance with regards to the experimental procedures described. In order to provide the highest level of accuracy, we replaced the “determined” with the “estimated” in the phrase: Total polyphenol contents of AJ and MJ were previously estimated by the Folin-Ciocalteu method and expressed as gallic acid equivalents (Line 121).
Line 107: What time over the day was the juice intake?. Indicate it could be important to avoid Chrono biological differences
Authors’ reply: We have included the sentence: The subjects consumed their daily dose after breakfast (L116).
Line 119: Have estimated the authors the polyphenols intake of the participants, using, for example the phenol explored platform?
Authors’ reply: Thank you for the important notion. We have additionally estimated the polyphenol intake by Serbian Food Composition Database, developed according to EuroFIR standards, that uses data on polyphenol content in foods from the on line accessible Phenol-Explorer database. This information is now included in the Materials and Methods section of the revised manuscript, Lines 142-143. As for the dietary intake, there were no significant differences between the studied groups for the estimated total polyphenol intake at the beginning of the study. These data were included in Table S1. The compliance with the instructed avoidance of the consumption of berries and some polyphenol-rich foods, was monitored by regular contact with members of the research team and unannounced 24h recall checks.
Line 144: Several characters are missing along the manuscript
Authors’ reply: Thank you, we put efforts to perform additional proofreading.
Line 166: Please, provide the primers used in the RT-PCR
Authors’ reply: For the genotyping of the selected SNPs, we have used commercially available, validated assays, and the full genotyping including the selection of variants has been described before (reference number 27, PMID: 33011673). The number of the assay, as given in the paper (L192-193), might provide all the relevant technical data for the assay on the website of the manufacturer, including context sequence for the probe within SNP position. As stated in the Methods section (2.5. Genotyping), the genotypes of all three SNPs were determined using TaqMan® Assays for SNP genotyping, designed and functionally tested by the manufacturer, Thermo Fisher Scientific. The manufacturer did not provide information on the sequence of primers, but on the context sequence within which the allele specific probes were designed. This and other details are provided online (https://www.thermofisher.com) for each assay used, under its product name: C___2575520_10 (for rs174576 polymorphism), C___2575513_10 (for rs174593 polymorphism) and C___2268923_10 (for rs174616 polymorphism). We trust in this form, we have provided sufficient information to allow reproducibility of the methodological approach, please see L188-193.
178: Why the waist and hip circumference, and its relation did not include in the analysis?. The waist/hip circumference is a best predictor of cardiovascular risk than the body weight or BMI.
Authors’ reply: We acknowledge the importance of waist circumference, hip circumference as well as the corresponding ratio as indicators of visceral obesity. In this study, we wanted to focus on the total obesity parameters, and thus analyzed BMI and total body weight as proxies. Follow-up analyses with waist circumference included as one of the crucial hallmarks of metabolic syndrome, are granted.
Line 196: Study include only overweight subject, no obese?. If then, please indicate the range of BMI
Authors’ reply: Thank you for this important comment. In this study we have included subjects with BMI more or equal to 25, meaning that a portion of the obese subjects has also been included (L372-374). To address important notion, we have added a term “subjects who are overweight and obese” where appropriate along the manuscript, and paid attention to avoid confusion, as our analyses included both subjects who were overweight, inclusive of those who were obese at the same time. We also indicated the BMI range (“BMI≥25”) accordingly.
Line 196: Could authors include ± SEM of age?
Authors’ reply: Thank you for this comment. We have added the SEM values of the study participant’s age in the text (Line 227).
Line 199: It is necessary indicated the methodology to determine the intake of PUFA and others FA in the diet
Authors’ reply: We determined the intake of main types of FAs (SFA, MUFA, and PUFA) as well as individual FAs in the study participants’ diet by using the Serbian Food Composition Database, standardized according to the EuroFIR requirements, that contains comprehensive data on nutritional values of foods and composited dishes. We specified that FA were calculated from 24-h recall dietary data using this database in the Material and Methods section of the revised manuscript (Lines 139-142).
Table 3: Please Body Weight, not Weight
Authors’ reply: Thank you for this suggestion. We have made the correction in the revised form of the manuscript.
Line 303: Please, define COMT
Authors’ reply: Thank you for this comment. We have provided the full gene name (Line 337).
Appendix Table A1: Please, check the data. Total number subjects = 62. Subjects rs174593 = 35, Subjects rs174616 = 48, Subjects rs174576 = 39; SUM = 122
Authors’ reply: The numbers refer to the number of subjects who are variant allele carriers within the three variant places. As a portion of subjects represent minor allele carriers for 2 or even 3 SNPs, the numbers overlap, adding to the number of 122.
Appendix Table A1: The total intake of fat (%TE) seem to be very high, corresponding to HFD
Authors’ reply: Thank you for the important notion. Our subjects really are with substantially high-fat intake, also mirrored in saliently high omega-6/omega-3 intake index of up to 17 (Table A1). We trust this notion emphasizes the need for the development of nutritional and dietary strategies in those, otherwise healthy, subjects.
Line 685: Please, review the format
Authors’ reply: Thank you for the suggestion the references were revised and corrected to match the format of the journal.
Round 2
Reviewer 1 Report
The first sentence of the title is positively answered in Table 2. There is a statistically strong relationship between changes in EPA and decreases in BMI and body weight in the total group. The second sentence of the title is more speculative. From Table 2, using an adjusted regression model (model M2), the statistical significance for EPA and polyphenol intake is lost. If the manuscript had focused on only the first sentence, then I would give the manuscript a higher ranking.
Since the AA/EPA ratio in these subjects was incredibly high at the beginning of the study, rather than exploring gene-polyphenol combinations, I believe simply reducing the AA/EPA ratio with various levels of omega-3 fatty acid supplementation would be more insightful since one would expect a dose-response effect on cytokine levels.
The primary enzyme associated with the FADS2 gene is delta-6-desaturase, I believe the authors would be better off looking for genetic variants of the FADS1 gene that affects the delta-5-desaturase enzyme that is a rate-limiting step in the production of EPA since it only EPA that was associated with weight loss. But that is water under the bridge.
I would suggest removing the last sentence of the abstract, and also rewriting the conclusion to state that increasing EPA intake may be a more suitable approach to reducing weight instead of the more speculative conclusion linking a gene variant in the FASD2 gene and chokeberry polyphenols for personalized nutrition to reduce obesity and its associated co-morbidities.
Reviewer 2 Report
Thanks to the authors for the new version of the manuscript and the kind replies to all the questions. In my opinion, the present form of the manuscript is suitable to be published by Nutrients.
